# Chemical Composition, Fatty Acid Profile, and Optimization of the Sacha Inchi (*Plukenetia volubilis* L.) Seed-Roasting Process Using Response Surface Methodology: Assessment of Oxidative Stability and Antioxidant Activity

**DOI:** 10.3390/foods12183405

**Published:** 2023-09-12

**Authors:** Nelsy Bocanegra Morales, Paula Galeano Garcia

**Affiliations:** Grupo de Investigación en Productos Naturales Amazónicos—GIPRONAZ, Facultad de Ciencias Básicas, Universidad de la Amazonia, Florencia 180001, Colombia; n.bocanegra@udla.edu.co

**Keywords:** sacha inchi, roasting, oxidative stability, antioxidant activity, fatty acids

## Abstract

This study aimed to optimize the roasting conditions for sacha inchi (*Plukenetia volubilis* L.) seeds using the central composite design (CCD) of the response surface methodology (RSM). The antioxidant activity and oxidation indicators (peroxide and TBA values) were assessed, along with the impact of roasting on the fatty acid profile and chemical characterization of the seeds using gas chromatography. The results demonstrated that roasting partially increased the indicators of lipid oxidation in the oil extracted from roasted seeds, as well as the antioxidant activity of the seeds. The optimal roasting conditions were determined using CCD and RSM, resulting in an optimized temperature of 134.28 °C and 18.84 min. The fatty acid contents were not significantly affected by the roasting intensity, whereas a higher presence of amino acids was found in the seeds roasted at 140 °C for 15 min. In conclusion, it is suggested that the optimal roasting conditions for enhancing amino acid presence, improving antioxidant activity, and maintaining oxidative stability in sacha inchi seeds fall within the temperature range of 134–140 °C and a roasting duration of 15–20 min.

## 1. Introduction

The Amazon region is home to a vast array of plant species, which are of great significance to global agriculture [1]. However, there are still numerous lesser-known fruit species that have potential economic value owing to their chemical properties. Sacha inchi (*Plukenetia volubilis* L.) is an oleaginous Amazonian fruit tree from the Euphorbiaceae family, commonly referred to as “wild peanut”, “Inca peanut”, “Inca inchi”, or “mountain peanut” [2,3]. Sacha inchi is widely distributed in South America, particularly in the Amazon River basin. Peru currently leads sacha inchi production and industry, with an annual seed production of approximately 1200 tons [4]. Nevertheless, other countries, such as Colombia, Ecuador, and Bolivia, have also begun to engage in agricultural and economic ventures [5]. In Colombia, sacha inchi cultivation has expanded to the southern regions of the country, notably the departments of Putumayo, Caquetá, Meta, Guaviare, and Cauca [6], showing promising economic potential and offering possibilities for replacing illicit crops in conflict-affected areas in Colombia [7]. 

*P. volubilis* cultivation has recently been implemented in various Asian countries, particularly in China, Cambodia, Thailand, and Laos [8]. Historical records pertaining to the use of sacha inchi primarily emanate from Peru, where ethnic groups have cultivated it for centuries as a source of sustenance and medicine [9]. Traditionally, roasted and salted seeds, or those coated with chocolate, are consumed. Furthermore, the seeds are pressed to obtain oil and ground to produce flour [10]. However, sacha inchi has predominantly served medicinal purposes, as evidenced by an ethnobotanical study conducted in San Martín, Peru [8,11]. Indigenous communities blend oil and ground seeds to create a cream for skin rejuvenation, alleviate muscular pain, and mitigate rheumatism. Additionally, oil and roasted seeds are utilized to manage cholesterol levels, promote cardiovascular health, and address gastrointestinal conditions [11]. Multiple studies have also demonstrated the bioactive properties of sacha inchi, endowing it with significant potential in the food, pharmaceutical, and cosmetic industries [12]. Presently, sacha inchi is commercially available in the form of oil, encapsulated oil, seeds, and protein powder, serving as a dietary supplement [8].

The chemical composition of sacha inchi varies depending on the plant part, with the seeds being of particular interest due to their high oil content (35–60%) and protein content (27%). The oil stands out because of its elevated levels of linolenic (ω-3) and linoleic (ω-6) acids, which account for approximately 45% and 35% of the total unsaturated fatty acids (UFAs), respectively. These fatty acids are renowned for their health benefits in preventing cardiovascular diseases, cancer, and hypertension [9,13,14]. Additionally, the presence of essential amino acids such as cysteine, tyrosine, threonine, and tryptophan [15] makes roasted sacha inchi seeds and sacha inchi oil highly valuable as functional foods, earning the title of “superfood” due to their exceptional nutritional value [16].

The roasting process of sacha inchi seeds plays a crucial role in snack production. This process involves subjecting the seeds to temperatures ranging between 80 and 120 °C, which enhances their flavor and aroma while eliminating undesirable tastes resulting from the presence of compounds such as tannins, saponins, and phytic acid [17,18]. This is also safer because sacha inchi contains appreciable amounts of alkaloids, saponins, and lectins, which may be toxic if consumed before cooking [18]. Nevertheless, it is important to note that thermal treatment affects both the chemical composition and nutritional properties. Cisneros et al. [19] conducted a study on sacha inchi oil and observed that roasting enhanced its oxidative stability and antioxidant capacity through the formation of phenolic compounds. Similar studies on cashews and hazelnuts have reported decreases in tocopherol and carotenoid content, as well as certain phenolic compounds, such as protocatechuic acid, phloretin-2-O-glucoside, catechin, and epicatechin, following roasting or drying [20,21]. Nonetheless, it is worth mentioning that the total phenolic content and antioxidant capacity tend to increase [22], indicating that roasting induces structural changes within the cellular matrix of food, potentially leading to the formation of novel phenolic compounds.

Furthermore, Bueno-Borges et al. [23] found that toasting the seeds at 160 °C for 15 min reduced the antinutrient content and increased the antioxidant activity of methanolic extracts of the seeds, while the oil oxidation indicators remained within acceptable quality limits. In another study, three types of cooking processes were analyzed: uncooked, roasted at 160 °C for 6 min, and boiled at 100 °C for 13 min. The results indicate that roasted sacha inchi is distinguishable by its high antioxidant content (polyphenol, flavonoid, and free radical scavenging activity) [18]. Based on these findings, the objective was to optimize the toasting conditions of the seeds using a central composite design (CCD) within a temperature range of 80–200 °C and a time range of 10–20 min. The present study aimed to evaluate the oxidative stability of oil and the antioxidant activity of methanolic seed extracts. Additionally, the profile of fatty acids in oil and the chemical composition of seeds at low (80 °C, 20 min), medium (140 °C, 15 min), and high (200 °C, 20 min) toasting were analyzed.

## 2. Materials and Methods

### 2.1. Collection and Toasting of Sacha Inchi Seeds

The encapsulated fruits of sacha inchi were donated by the Sacha Caquetá association in the municipality of Doncello (Caquetá, Colombia) when they were ripe and dry. Seeds were manually dehusked and toasted in a conventional oven. Response surface methodology (RSM) was used in the present study to optimize the toasting conditions. In the central composite design (CCD), the factors, levels, temperatures (80–200 °C) and times (10–20 min) were evaluated, with antioxidant activity and oxidation indicators (peroxide value and TBARS value) used as response variables. The selection of design levels was based on a literature review. Thirteen toasting conditions were applied with five repetitions at the central point (Table 1). After toasting, the seeds were cooled to room temperature and stored to prepare hydroalcoholic extracts.

### 2.2. Sacha Inchi Oil Extraction

The oil extraction process was conducted by cold pressing using a Mega KPD-30A hydraulic press, exerting a force of 30 tons. A total of 100 g of sacha inchi seeds was packed into fine mesh cloth bags to prevent any solid particles from contaminating the oil. An applied force of 450 kg/cm^2^ was used for oil extraction at room temperature. The extracted oil was meticulously collected using a Pasteur pipette, decanted, and stored in an amber glass container at −40 °C until subsequent analysis [19].

### 2.3. Assessment of Antioxidant Capacity

#### 2.3.1. Preparation of Hydroalcoholic Extracts

A total of 1 g of the sample was dissolved in 15 mL of a 70:30 (*v*/*v*) methanol (Merck, Rahway, NJ, USA)/water solution and stirred for 1 h at room temperature. The extract was then centrifuged at 4500 rpm for 15 min at room temperature. The supernatant was filtered and adjusted to a volume of 25 mL by using the same solvent. The extracts were stored in amber vials at 4 °C.

#### 2.3.2. ABTS Free Radical Scavenging Method

To prepare the ABTS (Merck) cationic radical, a solution of 7 mM ABTS and 2.45 mM potassium persulfate was mixed. After 24 h, the ABTS˙^+^ solution was diluted with phosphate-buffered saline (PBS) at pH 7.4, until an absorbance of 0.700 ± 0.020 at 734 nm was reached. For the analysis, 3 µL of the extract was mixed with 295 µL of the diluted ABTS˙^+^ solution, and the absorbance was measured at 734 nm after 30 min of reaction in the dark [24]. Antioxidant activity was determined using a calibration curve with ascorbic acid standards, and the results were expressed as µmol equivalents of ascorbic acid (Merck) per gram of sample (µmol EAA/g sample).

### 2.4. Thiobarbituric Acid (TBA) Value and Peroxide Value

The peroxide value was determined using the standard IDF 74A:1991 method [25,26]. This method quantifies the hydroperoxides formed by the primary oxidation of lipids, and the results are expressed in milliequivalents of oxygen per Kg of oil. The TBA value was determined using the methodology described previously by Zeb and Ullah [27] with some modifications. A weight of 0.05–0.5 g of oil was added to a volume of 3 mL with glacial acetic acid in a Falcon tube. Then, 3 mL of a thiobarbituric acid (TBA) (Sigma-Aldrich, St. Louis, MO, USA) solution (200 mg in 100 mL of glacial acetic acid (Sigma-Aldrich)) was added, and the mixture was placed in a water bath at 95 °C for 1 h. After cooling to room temperature, the intensity of coloration was measured at 532 nm using a spectrophotometer [28].

### 2.5. GC-FID Fatty Acid Profile

The fatty acid profiles of raw sacha inchi oil and oils from lightly roasted (80 °C, 20 min), moderately roasted (140 °C, 15 min), and highly roasted (200 °C, 20 min) samples were determined using gas chromatography following the Colombian standard method NTC 4967 [29]. Methyl esters of fatty acids were analyzed on an Agilent Technologies (Santa Clara, CA, USA) gas chromatograph (6890N) equipped with a split/splitless injector, a flame ionization detector (FID), and a DB-225 capillary column (30 m × 0.25 mm, 0.25 µm, Agilent Technologies). The injector and detector temperatures were set at 250 °C and 220 °C, respectively [29]. The oven was initially maintained at 75 °C and programmed using a time-based heating ramp to 220 °C at a rate of 5 °C/min. Helium was used as the carrier gas with an injection volume of 0.2 µL. The results are expressed as grams of fatty acids per 100 g of oil.

### 2.6. Characterization by Gas Chromatography–Mass Spectrometry (GC-MS)

Chloroform (Merck)/methanol (Merck)/water extracts (1:3:1 *v*/*v*) of raw sacha inchi seeds that were lightly roasted (80 °C, 20 min), moderately roasted (140 °C, 15 min), and highly roasted (200 °C, 20 min) were lyophilized. To the lyophilized samples, 20 µL of O-methoxylamine in pyridine (15 mg/mL) (Sigma-Aldrich) was added, followed by vortexing at 3200 rpm for 5 min and incubating in the dark at room temperature for 16 h. Sililation was performed by adding 20 µL of BSTFA with 1% TMS (Sigma-Aldrich), followed by vortexing for 5 min and incubating at 70 °C for 1 h. Finally, the samples were cooled to room temperature for 30 min, and 180 µL of heptane (Merck) was added. The mixture was vortexed for 10 min at 3200 rpm.

GC-MS analysis was performed using an Agilent Technologies 7890B gas chromatograph coupled with an Agilent Technologies GC/Q-TOF 7250 time-of-flight mass spectrometer. The system was equipped with a split/splitless injection port (250 °C, split ratio of 50) and an Agilent Technologies 7693A automatic injector. Electron ionization (EI) was operated at 70 eV. An Agilent Technologies J&W HP-5MS column (30 m, 0.25 mm, 0.25 µm) was used with helium as the carrier gas at a constant flow of 0.7 mL/min. The oven temperature was programmed to increase from 60 °C (1 min) to 325 °C at a rate of 10 °C/min. The transfer line, filament, and quadrupole temperatures were maintained at 280 °C, 230 °C, and 150 °C, respectively. Mass spectrometry detection was performed in the range of 50–600 *m*/*z* at a speed of 5 spectra/min.

## 3. Results and Discussion 

### 3.1. The Effect of Roasting on the Organoleptic Properties of Sacha Inchi Seeds

The roasting process of sacha inchi seeds induces notable modifications in their physical attributes, leading to discernible changes in color, flavor, and aroma [15,19]. Initially, the untreated seeds exhibit a creamy hue, which undergoes a subtle transition to a light brown shade as the roasting temperature gradually increases within the range of 80–140 °C. Upon reaching higher roasting temperatures (170–200 °C), the seeds acquired a more pronounced dark brown coloration (Figure 1). Concurrently, elevated roasting temperature intensifies the aroma reminiscent of peanuts, while the typical astringent taste of beans diminishes. However, it is worth noting that beyond the threshold of 200 °C, the seeds tended to undergo carbonization, imparting an undesirable bitter taste. Similarly, changes in color and odor have been observed in oil [16,19].

### 3.2. The Effect of Roasting on the Antioxidant Activity in Sacha Inchi Seeds

To optimize the temperature and roasting time parameters for sacha inchi seeds, an experimental design using response surface methodology was employed. The objective of this study was to investigate the impact of these variables on both the antioxidant activity of the seeds and the indicators of oxidation in sacha inchi oil.

Figure 2A presents a response surface plot depicting the intricate relationship between the independent variables (temperature and time) and the response variable represented by antioxidant activity (µmol EAA/g seed). The antioxidative activity response attained its highest value at a temperature of 200 °C and a toasting duration of 20 min. The experimental results were fitted to a well-fitting quadratic model with an insignificant lack of fit (*p* > 0.05). The ANOVA yielded a high coefficient of determination (R^2^) of 0.9947, with a coefficient of variability (%CV) of 0.28. Both temperature and time variables were found to be statistically significant (*p* < 0.05), with temperature exhibiting a particularly high level of significance (*p* < 0.0001). The temperature range of 80–200 °C and the time range of 10–20 min exert a statistically significant (95%) influence on antioxidative activity. Furthermore, they exhibit positive effects, wherein an enhancement in their values leads to increased antioxidant activity (Figure 2B). Furthermore, the graph demonstrates that lower temperatures and shorter roasting times correspond to decreased antioxidant activity (blue color), whereas higher temperatures and longer roasting times result in increased antioxidant activity (red color) (Figure 2A).

The antioxidant activity of sacha inchi seeds roasted at different temperatures and exposure times was assessed using the ABTS method. The results demonstrate an increase in antioxidant activity following roasting, ranging from 2606.69 µmol EAA/g seed (at 80 °C for 10 min) to 2905.99 µmol EAA/g seeds (at 200 °C for 20 min) (Table 1). Consistent with these findings, Bueno-Borges (2018) reported that the highest roasting temperature (160 °C for 15 min) resulted in greater antioxidant activity in sacha inchi seeds.

In nature, the presence of phenolic compounds, either free or covalently bound to other groups or molecules, has been documented [30]. Roasting disrupts these bonds, leading to an increase in antioxidant activity. Additionally, the Maillard reaction during roasting generates compounds with antioxidant properties, which likely contribute to the observed increase in antioxidant activity at specific roasting temperatures [23,31]. This heat-induced enhancement of antioxidant activity has also been reported for other food products [32,33].

Similar observations have been made in studies involving peanut oil (*Arachis hypogaea* L.) [34] and sesame seeds (*Sesamum indicum* L.), wherein the formation of dicarbonyl compounds, acrylamide, and 5-hydroxymethylfurfural was detected [35]. Furthermore, roasted sunflower seeds have been found to contain volatile compounds, including aldehydes, alcohols, ketones, pyrazines, and furans, resulting from Maillard and lipid oxidation reactions [36]. These compounds exhibit secondary oxidation with potential reducing properties [31]. Notably, certain Maillard products have shown promising anticancer and antimicrobial properties [37], thereby broadening their potential benefits beyond enhancing the antioxidant capacity of food.

### 3.3. The Effect of Roasting on Sacha Inchi Oil Oxidation

Sacha inchi oils were obtained by cold pressing roasted seeds at various temperatures and exposure times (Table 1). Figure 3 presents the response surface plot, demonstrating the interplay between the independent variables (temperature and time) and the response variables, namely the peroxide value (mEqO_2_/kg) and TBA value. The peroxide value showed a satisfactory fit with a quadratic model (R^2^ = 0.9574, %CV = 6.65) (Figure 3A), while the TBA value exhibited a linear relationship with an acceptable fit (R^2^ = 0.9017, %CV = 7.68) (Figure 3B). Both fitted models present a non-significant lack of fit (*p* > 0.05). The ANOVA results revealed a high significance for temperature (*p* < 0.0001), whereas time had no significant influence on the response variables of oxidative stability. The Pareto charts (Figure 3C,D) illustrate that temperature (80–200 °C) had a significant effect (95%), whereas time (10–20 min) did not exhibit a significant effect (95%) on the variables of oxidative stability. However, both temperature and time have positive effects, meaning that an increase in their values leads to an increase in the peroxide and thiobarbituric acid (TBA) values. Figure 3A,B indicates that lower roasting temperatures corresponded to decreased peroxide and TBA values (depicted in blue), whereas higher temperatures and longer exposure times led to increased values (depicted in red).

Similar to antioxidant activity (Table 1), the oxidation indicators demonstrated an increasing trend with increasing temperature. TBA values ranged from 0.016 (80 °C, 10 min) to 0.035 (200 °C, 20 min), suggesting that roasting affects lipid oxidation in sacha inchi oils, leading to the formation of secondary oxidation products, primarily aldehydes [27]. The reaction between 2-thiobarbituric acid (TBA) and malondialdehyde (MDA), the primary marker of lipid peroxidation, yields a colored complex that can be measured spectrophotometrically [27].

The peroxide value increased from 1.91 mEqO_2_/kg of oil (80 °C, 10 min) to 3.25 mEqO_2_/kg of oil (200 °C, 20 min) (Table 1), indicating an increase in primary oxidation products, which is attributed to the accumulation of hydroperoxides resulting from the attack of free radicals on unsaturated fatty acids, such as linolenic acid [32,34], which is present in high concentrations in sacha inchi oils [19]. The increase in oxidation indicators with the intensity of roasting, linked to high linolenic acid content, has also been reported for flaxseed oil [38] and other seeds [34,39].

The increase in antioxidant capacity and oxidative stability can be attributed to the newly formed Maillard products and the formation of phenolic compounds [32,39]. Furthermore, melanoidins produced as a result of the Maillard reaction improve oxidative stability and impart distinct color and flavor characteristics to the oil [39]. Despite the increase in peroxide values observed at higher roasting temperatures, they remained below the permissible limits for high-quality oil (below 10 meqO_2_/kg) [40]. Thus, the temperatures and exposure times assessed for sacha inchi seed roasting in DCC did not have a negative impact on oxidative stability. Oil is likely capable of resisting oxidative deterioration, lipid hydrolysis, and microbial degradation by eliminating moisture content through thermal treatment. Roasting, along with encapsulation and the addition of natural antioxidants, is considered one of the methods to reduce the peroxidation of polyunsaturated fatty acids [3,15,41,42].

### 3.4. Fatty Acid Profiles of Sacha Inchi Oil

The fatty acid composition of sacha inchi oil subjected to different thermal treatments is presented in Table 2. The most abundant fatty acids are linolenic acid (43.3–44.3 g/100 g), linoleic acid (33.7–35.0 g/100 g), oleic acid (11.3–12.4 g/100 g), palmitic acid (4.59–4.91 g/100 g), stearic acid (3.36–3.68 g/100 g), docosanoic acid (0.02–0.07 g/100 g), and eicosanoic acid (0.114–0.134 g/100 g). Saturated fatty acids (palmitic acid, stearic acid, eicosanoic acid, and docosanoic acid) are present in low amounts, accounting for 9% of the total, while unsaturated fatty acids (linolenic acid, linoleic acid, and oleic acid) are highly abundant, accounting for 91% of the total. These findings indicate that sacha inchi oil could be a favorable dietary option owing to its low saturated fatty acid content, which is consistent with previous studies by Cisneros et al. [19], Kim and Joo [14], and Hamaker et al. [43]. Notably, even at the high toasting temperature of 200 °C for 20 min, the fatty acid profile of sacha inchi oil remained largely unaffected, which is consistent with the findings reported by Cineros et al. [19] at lower toasting temperatures (102 °C for 10 min), as well as similar observations in the fatty acid profiles of oils derived from other oilseeds [44,45]. 

Keawkim and Na Jom [46] discovered that thermal treatment significantly increases the levels of free fatty acids in both germinated and non-germinated sacha inchi seeds. This effect is likely attributed to the heat-induced degradation of lipase activity, which tends to decrease at higher temperatures and prolonged exposure times. Consequently, damaged cells containing hydrolytic enzymes may contribute to elevated levels of free fatty acids following toasting [46].

According to Fanali et al. [13], sacha inchi oil offers more beneficial nutraceuticals because of its considerably higher ω6 content than common oilseeds such as canola, flaxseed, sunflower, and soybean oil. Moreover, the well-balanced ratio of ω3 to ω6 fatty acids in sacha inchi oil, which approaches the ideal range of 1:4 to 1:10, exerts a significant hypocholesterolemic effect. Additionally, it contributes to the prevention of cardiovascular and inflammatory diseases while reducing the risk of cancer [14,15].

### 3.5. GC-MS Profile of Sacha Inchi Seeds

Dynamic changes in unroasted and roasted sacha inchi seeds at different temperatures and times are presented in Table 3. Thirty-one identified metabolites were classified into five groups. Group I contained eight amino acids, group II contained three fatty acids, group III contained eight organic acids, group IV contained two alcohols, and group V contained ten sugars. In the GC-MS analysis, 24 compounds were identified in unroasted sacha inchi seeds, 25 in lightly toasted seeds (80 °C, 20 min), 24 in moderately toasted seeds (140 °C, 15 min), and 21 in highly roasted seeds (200 °C, 20 min), including amino acids, organic acids, fatty acids, alcohols, and sugars (Table 3). Of the thirty-one identified compounds, sixteen were found to be common in all four samples, indicating that the degree of toasting did not affect the presence of these components in the seeds. 

Of 24 identified compounds, 20 have been previously reported in both sacha inchi seeds and oil. An essential amino acid (L-valine) and two essential fatty acids (linolenic acid and linoleic acid) were found in both roasted and unroasted seeds. The presence of essential compounds, which cannot be synthesized by the body and, therefore, must be obtained through diet, makes sacha inchi a “superfood” with significant benefits for human health [47,48]. L-valine was detected in all heat treatments, whereas L-leucine was only present in low-temperature roasting. Wang et al. [15] have also identified other essential amino acids in sacha inchi seeds, including lysine, isoleucine, tyrosine, and threonine. Generally, there was a higher concentration of amino acids during medium roasting, but their levels decreased at 200 °C, which aligns with Keawkim and Na Jom’s [46] finding of a slight reduction in amino acid content at 180 °C during prolonged roasting. The non-essential amino acids identified included L-alanine, L-serine, L-glutamic acid, and L-5-oxoproline. L-5-oxoproline, present in all heat treatments, is a cyclic amino acid formed through the dehydration of glutamate [49] or heating of L-glutamic acid between 160 and 180 °C [50]. This may explain the disappearance of L-glutamic acid at 180 °C and the presence of L-5-oxoproline at higher temperatures. Furthermore, the non-protein amino acid 4-aminobutanoic acid and gamma-aminobutyric acid (GABA) were detected at 140 °C. GABA is a powerful bioactive compound that is crucial for brain function, acting as an inhibitory neurotransmitter in the central nervous system and exhibiting hypotensive effects [51].

It has been demonstrated that the increase in free amino acids during roasting is attributed to the denaturation of certain proteins [21,47]. The findings from this study revealed a higher presence of amino acids in medium-roasted samples, followed by lightly roasted samples, and a decrease in high-roasted samples. The decrease in certain amino acids at high roasting temperatures can be attributed to their oxidation and the formation of advanced glycation end-products, such as carboxymethyllysine (CML) and carboxyethyllysine (CEL), resulting from the reaction of amino acids with reducing sugars. Consequently, high temperatures can cause structural damage and modification of amino acid residues [52]. Therefore, the cooking and roasting processes generate Maillard reactions (glycation), leading to various pathways and interactions, including caramelization, Strecker degradation, and decomposition of sulfur-containing amino acids, depending on the roasting intensity [53]. Thus, a roasting temperature of 140 °C with a 15 min exposure improved the amino acid content compared to the other treatments.

Similarly, eight sugars were found in all the heat treatments. The reducing sugars D-glucose and D-fructose, sucrose, and myoinositol have been reported in other studies on sacha inchi seeds [46,54,55]. These sugars may be responsible for the formation of Maillard reaction products during toasting, contributing to the characteristic color and flavor of the toasted seeds. Sugars and amino acids found in toasted seeds are essential substrates for the formation of color, flavor, and aroma during the Maillard process [46]. Furthermore, these compounds also contribute to the formation of Maillard reaction products that are associated with improved antioxidant properties and oxidative stability [37]. Previous studies have identified some Maillard reaction products in sacha inchi seeds, such as pyrazines, which can be formed through the sugar–amino acid system [46]. Similar effects of roasting have been observed in other fruits, further supporting the presence of Maillard products contributing to antioxidant properties [32].

In addition to amino acids and sugars, organic acids have also been detected in sacha inchi seeds. Lactic acid and glycolic acid were present in all treatments and could be formed through the conversion of sucrose and fructose into their intermediates, 1,2-endiol and 2,3-endiol, respectively (Appendix A) [56,57]. These organic acids are involved in Maillard reactions and may contribute to the characteristic flavor and aroma of toasted seeds. Malic acid was detected in all treatments, whereas 2-hydroxyglutaric acid was found only in seeds toasted at 80 °C, and 2-butenedioic acid and butanedioic acid were identified in untoasted seeds. For fatty acids, thermal treatment did not affect the content of sacha inchi oil, and the results indicated that the degree of toasting did not influence their presence in the seeds.

## 4. Conclusions

The roasting process of sacha inchi seeds led to a slight increase in the oxidation indicators in the oil and the antioxidant capacity of the seeds. The optimal roasting conditions were determined using a central composite design and response surface methodology, resulting in a recommended temperature of 134.28 °C and a duration of 18.84 min. These conditions were based on peroxide value, TBA value, and antioxidant activity. Interestingly, roasting intensity did not significantly affect the content of polyunsaturated fatty acids, such as linolenic acid and linoleic acid. Among the amino acids, the highest level was observed in seeds roasted at 140 °C, followed by those roasted at 80 °C, and unroasted seeds. However, at a roasting temperature of 200 °C, both amino acids and organic acids decreased. Notably, the sugar content showed minimal changes with varying roasting intensity. Therefore, it can be concluded that the optimal roasting conditions to enhance the presence of amino acids, improve antioxidant activity, and maintain oxidative stability in sacha inchi seeds fall within the range of 134–140 °C and 15–20 min.

## Figures and Tables

**Figure 1 foods-12-03405-f001:**
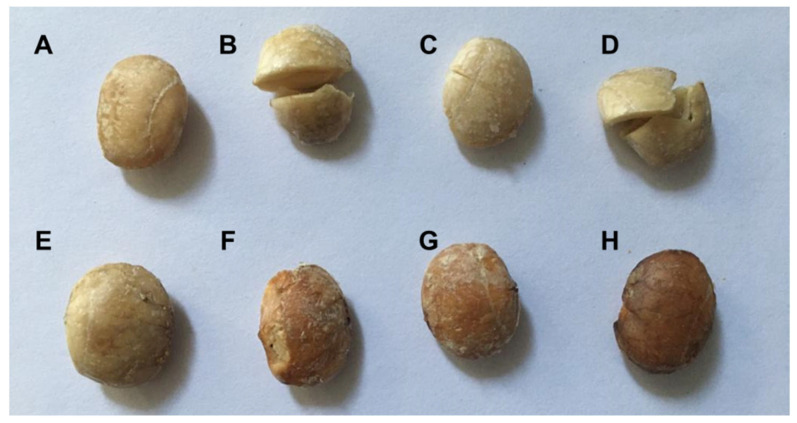
The effect of roasting on the physical appearance of the sacha inchi seeds in different heat treatments: unroasted (**A**); 80 °C, 10 min (**B**); 80 °C, 20 min (**C**); 110 °C, 15 min (**D**); 140 °C, 15 min (**E**); 170 °C 15 min (**F**); 200 °C, 10 min (**G**); 200 °C, 20 min (**H**).

**Figure 2 foods-12-03405-f002:**
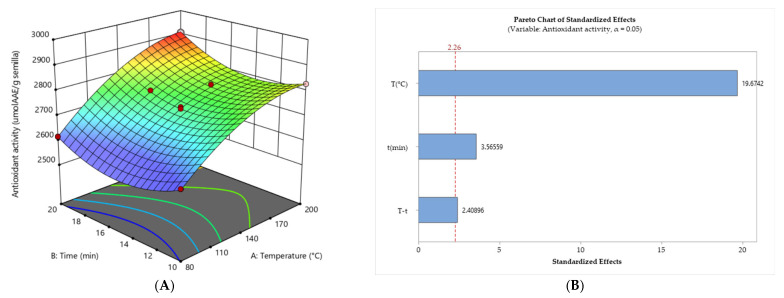
(**A**) The 3D response surface plot illustrates the impact of roasting temperature and time on the antioxidant activity of sacha inchi seeds. (**B**) Pareto chart showing the standardized effect of independent variables and their interaction on the antioxidant activity.

**Figure 3 foods-12-03405-f003:**
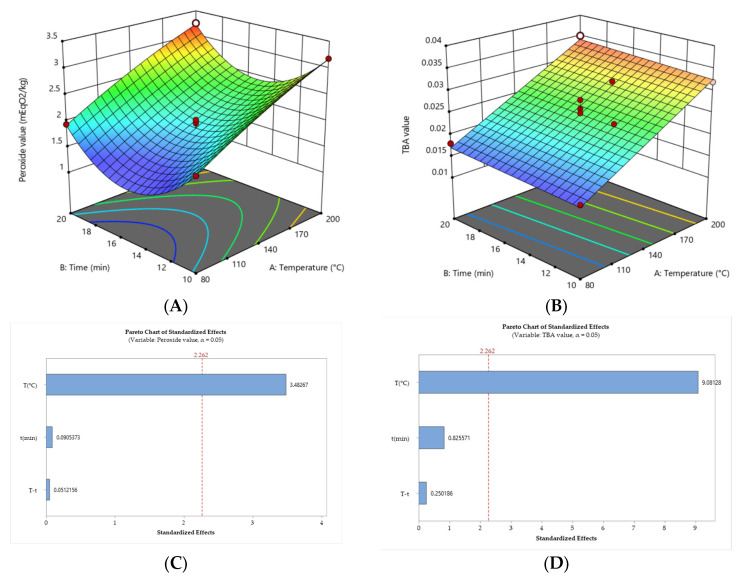
The 3D response surface plot depicts the relationship between time and temperature and their effect on two parameters: (**A**) peroxide and (**B**) TBA value. Pareto chart showing the standardized effect of independent variables and their interaction on the oxidation indicators, (**C**) peroxide, and (**D**) TBA value.

**Table 1 foods-12-03405-t001:** The central composite design and responses of the dependent variables to the sacha inchi roast seeds and oils.

Temperature (°C)	Time (min)	Roasted Seed Extracts	Roasted Seed Oils	
		ABTS ^a^	Peroxide Value ^b^	TBA Value
200	10	2828.37	3.18	0.032
80	20	2616.59	1.94	0.018
80	10	2606.69	1.91	0.016
140	15	2736.72	2.02	0.024
140	12.5	2727.10	1.98	0.025
140	15	2728.39	2.03	0.022
140	17.5	2764.69	1.93	0.022
170	15	2788.67	2.05	0.03
200	20	2905.99	3.25	0.035
140	15	2724.51	1.97	0.026
110	15	2637.58	1.44	0.019
140	15	2729.56	1.96	0.028
140	15	2728.15	1.72	0.025

^a^ Values are expressed as µmol EAA/g grain. ^b^ Values are expressed as mEqO_2_/kg (milliequivalents of oxygen per kilogram of seed).

**Table 2 foods-12-03405-t002:** Changes in the fatty acid composition of unroasted and roasted sacha inchi seeds during thermal treatments.

	Oil Sample			
Fatty Acid	Unroasted Seeds	Lightly Roasted Seeds (80 °C, 20 min)	Medium Roasted Seeds (140 °C, 15 min)	Highly Roasted Seeds (200 °C, 20 min)
Palmitic acid (C16:0)	4.74 ± 0.05 *	4.59 ± 0.05	4.60 ± 0.05	4.91 ± 0.05
Stearic acid (C18:0)	3.36 ± 0.04	3.37 ± 0.04	3.47 ± 0.04	3.68 ± 0.04
Eicosanoic acid (C20:0)	0.117 ± 0.001	0.120 ± 0.001	0.114 ± 0.001	0.134 ± 0.001
Docosanoic acid (C22:0)	0.04 ± 0.01	0.02 ± 0.01	0.03 ± 0.01	0.07 ± 0.01
Oleic acid (C18:1)	11.8 ± 0.1	11.3 ± 0.1	11.5 ± 0.1	12.4 ± 0.1
Linoleic acid (C18:2)	33.7 ± 0.4	33.8 ± 0.4	35 ± 0.4	34.5 ± 0.4
α-Linolenic acid (C18:3)	43.3 ± 0.5	43.8 ± 0.5	44.3 ± 0.5	43.2 ± 0.5

* The values are expressed in (g/100 g) of oil.

**Table 3 foods-12-03405-t003:** Chemical composition of sacha inchi seeds with different heat treatments.

		Roasted Seed			
Groups	Compounds	From Unroasted Seeds	Lightly Roasted Seeds (80 °C, 20 min)	Medium Roasted Seeds (140 °C, 15 min)	Highly Roasted Seeds (200 °C, 20 min)
Amino acids	L-Valine	+	+	+	+
	L-Alanine	+	ND *	+	+
	L-Aspartic acid	ND	+	+	ND
	L-Leucine	ND	+	ND	ND
	L-Serine	+	+	+	ND
	L- Glutamic acid	ND	+	+	ND
	4-Aminobutanoic acid	ND	ND	+	ND
	L-5-Oxoproline	+	+	+	+
Fatty acids	Linoleic acid	+	+	+	+
	α-Linolenic acid	+	+	+	+
	Palmitic acid	+	+	+	+
Organic acids	2-Hydroxyglutaric acid	ND	+	ND	ND
	Glyceric acid	+	+	ND	+
	Butanoic acid	+	ND	ND	ND
	Malic acid	+	+	+	+
	2-Butenedioic acid	+	ND	ND	ND
	Butanedioic acid	+	ND	ND	ND
	Lactic acid	+	+	+	+
	Glycolic Acid	+	+	+	+
Alcohols	2,3-Butanediol	+	+	+	+
	Glycerol	+	+	+	+
Sugars	D-Fructose	+	+	+	+
	Meso-erythritol	+	+	+	+
	D-Arabinose	+	+	+	+
	L-(-)-Arabitol	+	+	+	+
	D-Glucose	+	+	+	+
	D-Mannitol	+	+	+	+
	Scyllo-inositol	+	+	+	+
	Myo-Inositol	+	+	+	+
	D-Chiro-Inositol	ND	+	ND	ND
	Sacarose	ND	+	+	+

* ND: not detected. The highlighted compounds were not previously reported in sacha inchi.

## Data Availability

The datasets generated for this study are available on request to the corresponding author.

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
