# Peer review of "Chemical Composition, Fatty Acid Profile, and Optimization of the Sacha Inchi (Plukenetia volubilis L.) Seed-Roasting Process Using Response Surface Methodology: Assessment of Oxidative Stability and Antioxidant Activity"

_foods, 2023, doi:10.3390/foods12183405_

Round 1

Reviewer 1 Report

Manuscript ID: foods-2492768

Title: Chemical Composition, Fatty Acid Profile, and Optimization of Sacha Inchi (Plukenetia volubilis L.) Seed Roasting Process Using Response Surface Methodology: Assessment of Oxidative Stability and Antioxidant Activity.

Authors: Nelsy Bocanegra; Paula Galeano

Review of the manuscript

Comments

The manuscript presented for revision is interesting. This work is well organized and scientifically sound.  However, I have some minor comments: 

1.Introduction

Because many potential readers know little about Sacha Inchi (Plukenetia volubilis L.). Therefore, I suggest adding some information about this raw material - e.g. what is the production volume in South American countries, what is the traditional use of this raw material, what are the processing methods...

2.Materials and Methods

·        Please describe in detail the oil extraction process (cold pressing):

- temperature of the raw material poured into the hydraulic press

- the temperature of the oil leaving the press

- how the oil was purified (was it filtered, centrifuged, sedimented and decanted? to remove physical impurities)

·        Please also specify the weight of one batch of pressed raw material for testing. The capacity of the hydraulic press used was very high, usually much smaller presses are used for laboratory tests.

·        In the discussion of the results, the Authors describe the organoleptic characteristics of Sacha Inchi, but there is no information about this analysis - how many people made the assessment, what was the scale, what was assessed. Please complete it.

3.Results and discussion

·        The assessment of oxidative stability can be carried out by many methods. In this study, it would be advisable to use the Rancimat test (determination of oxidation induction time). I suggest using it in future works.

·        The Authors state that (line 254): "The increase in antioxidant capacity and oxidative stability can be attributed to the newly formed Maillard products and formation of phenolic compounds [27,35]". Many authors confirm that the oxidative stability of roasted seed oil is related to the formation of phenolic compounds. Why was such a simple study not carried out - the content of total phenolic compounds. This would add value to this peer-reviewed work. Without this designation, some conclusions are more guesswork, speculation...

4.Conclusions not “Conclusiones”

Author Response

We express our sincere appreciation to the reviewer for dedicating their time and expertise in evaluating the manuscript. Their suggestions and comments were of immense value. In response to their constructive feedback, we diligently addressed and augmented the introduction to encompass comprehensive insights into the production, utilization, and treatment of sacha inchi raw material in Latin America. We believe that supplementary information enriches the scholarly content pertaining to sacha inchi. Kindly review the sections highlighted on pages 1 and 2 of the manuscript.

Furthermore, we have meticulously elucidated the methodology employed for the extraction of sacha inchi oil, adhering to rigorous scientific rigor. We kindly request that the reviewer thoroughly evaluate the highlighted text on page 3 of the manuscript to ensure clarity.

Concerning organoleptic analysis, we acknowledge that due to practical constraints, it was not subjected to formal evaluation by an expert panel. Instead, the findings are rooted in the perceptions of our research group, which comprised eight individuals who diligently observed and recorded alterations in color, taste, and odor during the toasting process. We thank the reviewer for highlighting this limitation, and we acknowledge the importance of adopting a more structured approach involving a panel of trained evaluators in future investigations. We also agree with the reviewer's valuable suggestion to employ the Rancimat test to evaluate oxidative stability, and we commit to incorporating this assessment in our forthcoming research endeavors.

Moreover, we share the reviewer's perspective on the significance of analyzing phenolic compounds, which would undoubtedly augment our study's scientific merit and relevance. It would be interesting to perform this analysis; however, we did not have access to the sample. Initially, it was not conducted as existing literature already reports a positive correlation between antioxidant activity and phenolic content with toasting in sacha inchi seeds. We have included relevant references for your consideration.

  1. Keawkim, K.; Na Jom, K. Metabolomics and Flavoromics Analysis of Chemical Constituent Changes during Roasting of Germinated Sacha Inchi (Plukenetia Volubilis L.). Food Chem X 2022, 15, doi:10.1016/j.fochx.2022.100399.
  2. Bueno-Borges, L.B.; Sartim, M.A.; Gil, C.C.; Sampaio, S.V.; Rodrigues, P.H.V.; Regitano-d’Arce, M.A.B. Sacha Inchi Seeds from Sub-Tropical Cultivation: Effects of Roasting on Antinutrients, Antioxidant Capacity and Oxidative Stability. J Food Sci Technol 2018, 55, 4159–4166, doi:10.1007/s13197-018-3345-1.
  3. Cisneros, F.H.; Paredes, D.; Arana, A.; Cisneros-Zevallos, L. Chemical Composition, Oxidative Stability and Antioxidant Capacity of Oil Extracted from Roasted Seeds of Sacha-Inchi (Plukenetia Volubilis L.). J Agric Food Chem 2014, 62, 5191–5197, doi:10.1021/jf500936

Reviewer 2 Report

The manuscript "Chemical Composition, Fatty Acid Profile, and Optimization of Sacha Inchi (Plukenetia volubilis L.) Seed Roasting Process Using Response Surface Methodology: Assessment of Oxidative Stability and Antioxidant Activity" submitted to Foods journal as Research Article is interesting because optimizing the process of food processing is always a challenge and an open question, especially as regards the quality of food. Oxidative Stability and Antioxidant Activity in the same trend monitoring is worthwhile and interesting.

However, the manuscript should be taking into account several comments and recommendations as follow:

- The affiliations of both authors are the same, with no difference in being separate

- Line 89 – “TBAR value” should be TBARS value

- What is the reason for the significantly higher content of Docosanoic acid in Lightly roasted seeds

- several technical gaps occurred, such as missing bracket after Docosanoic acid in Table 2

- Table 2 – may be conducting Anova to establish the significant differences be possible.

- Why RSM was not applied to all the assays?

- Table 3 – “Fitty acid”, please correct it

- Why the chemical composition of sacha inchi seeds with different heat treatments is only qualitative? This is not enough for comprehensive research. No efficient comparison is possible. Please provide more data and if possible values.

- The authors should point out better the novelty for the auditory.

Author Response

We express our gratitude to the referee for dedicating time to carefully review the manuscript. These valuable suggestions are important. We apologize for typographical errors and have corrected the necessary changes accordingly.

Regarding the observation of docosanoic acid in slightly toasted seeds, we concur with the referee's observations. However, due to limited information on the behavior of docosanoic acid in toasting processes and considering its low nutritional value and presence at very low concentrations, we have been unable to provide a definitive explanation for this behavior. Therefore, it would be interesting to explore the behavior of docosanoic acid during thermal treatment in future studies.

In response to the suggestion in Table 2, we acknowledge that the GC-FID technique was validated by ISO/IEC 17025:2017, making the analyses costly. As a result, we opted to apply analysis of the fatty acid profile to a single sample, making it unfeasible to conduct an analysis of variance.

We appreciate the referee's suggestion for the quantitative analysis of the chemical composition of the seed. While we agree that such an analysis would enable a more comprehensive comparison between thermal treatments, our current focus was to establish the metabolite profiles in the different thermal treatments to ensure accuracy. Moreover, the necessary standards for conducting quantification at this stage are lacking. Nevertheless, the quantitative analysis of metabolites during the sacha inchi seed toasting process would be a relevant avenue for future research.

Furthermore, we would like to clarify that Response Surface Methodology (RSM) was not applied to all samples because of the associated costs of GC techniques. Instead, we believe that the chosen temperatures of the central composite design (80 °C, 140 °C, and 200 °C) for the fatty acid profile and chemical composition allowed us to effectively observe the toasting effect on sacha inchi seeds.